# Acute Effects of a Short Bout of Physical Activity on Cognitive Function in Sport Students

**DOI:** 10.3390/ijerph17103678

**Published:** 2020-05-23

**Authors:** Martin Niedermeier, Elisabeth M. Weiss, Lisa Steidl-Müller, Martin Burtscher, Martin Kopp

**Affiliations:** 1Department of Sport Science, University of Innsbruck, 6020 Innsbruck, Austria; Lisa.Steidl-Mueller@uibk.ac.at (L.S.-M.); Martin.Burtscher@uibk.ac.at (M.B.); Martin.Kopp@uibk.ac.at (M.K.); 2Department of Psychology, University of Innsbruck, 6020 Innsbruck, Austria; Elisabeth.Weiss@uibk.ac.at

**Keywords:** attention, cognition, exercise, cognitive functions, sedentary

## Abstract

Physical activity is a promising intervention to restore cognitive function after prolonged sedentary periods. However, little is known about the effect of short physical activity bouts on cognition especially among individuals that are used to physical activity. Therefore, the goal of the present study was to assess the impact of a single ten-minute physical activity bout on the cognitive domain of visual attention compared to sedentary behavior in a population of physically active sport students. Using a randomized controlled design, 51 healthy and physically active sport students [mean age: 22.3 (SD: 2.0) years, 33.3% female] were allocated to one of the following interventions in the break of a two-hour study course: physical activity group (running for ten minutes) and sedentary control group. Visual attention was measured post-intervention using a modified trail making test. Pre-, post-, and 30 min after intervention, perceived attention, and affective states were measured. Between-group comparisons were used to analyze whether visual attention and/or changes in perceived attention or affective states differed between groups. The physical activity group showed significantly higher visual attention post-intervention compared with the sedentary control group, *p* = 0.003, *d* = 0.89. Perceived attention, *p* = 0.006, *d* = 0.87, and arousal, *p* < 0.001, *d* = 1.68, showed a significantly larger pre- and post-intervention increase in the physical activity group compared with the sedentary control group, which was not evident 30 min after intervention. A single ten-minute running intervention in study breaks might help to restore the basal visual attentional domain of cognition after prolonged sedentary periods more effectively compared with common sedentary behavior in breaks between study lessons.

## 1. Introduction

In industrialized countries, a sedentary lifestyle, defined as a lack of physical activity, can be considered as predominant and increases the risk of adverse health outcomes including overall mortality [1], cardio-metabolic diseases [2], musculoskeletal diseases [3], and mental health disorders [4]. According to the World Health Organization, more than 50% of adults aged 18–64 years are considered as being insufficiently physically active in Austria [5] and about 70% of American adults are below the recommended physical activity level [6]. It is estimated that around 60% of the total waking time is conducted in a sedentary position in the UK population [7]. Especially for office workers, 65–70% of their working hours are spent sedentary [7], but sedentary behavior is existent in all main domains of living including travelling (e.g., time spent in the car) or during leisure time [6,8]. For example, college students spend as much as 30 h per week with sedentary activities such as reading, studying, using the computer, or watching TV [6]. However, a single prolonged period of sitting may negatively impact cognitive function [9]. Some aspects of cognition are impaired already after two hours of laboratory-based sitting computer work [10]. Since the productivity of (office) workers and students might decline due to sedentary-induced impaired cognitive function, effective restorative interventions are needed.

One promising intervention to restore cognitive function is a single bout of physical activity [11]. The integration of physical activity bouts in a working day can be achieved on several ways. In a consensus statement regarding the avoidance of prolonged sedentary periods, the authors recommend using sit-stand desks or short active standing breaks [7]. Furthermore, the authors recommend “accumulating 2 h/day of standing and light activity (light walking) during working hours” [7] (p. 1357). Speaking practically, an appropriate way to achieve this amount of physical activity might be to include short bouts (ten min and shorter) of physical activity. Thereby, a short break between meetings on a working day (for office workers) or the time between study lessons (for students) might be used to restore cognitive functions by a physical activity bout. 

There are at least four important aspects to consider when focusing on effects of acute physical activity bouts on cognitive functions. Firstly, an important distinction must be made regarding the timing of assessment of cognitive functions in acute single bouts of physical activity: When cognitive functions are assessed during the physical activity bout, effects are predominately negative, i.e., increased response time to a cognitive task [12]. Following a single bout of physical activity after a delay (i.e., at least one minute between end of physical activity and assessment of cognitive functions), meta-analytic evidence is available for small-to-moderate-sized beneficial effects on cognitive functions, especially in the executive domain [11,12,13]. Secondly, the intensity of the physical activity bout might produce different effects. Although discussed controversially and connected to other characteristics of the physical activity bout (e.g., duration), the intensity–cognitive improvements relationship might follow an inverted-U function, where the largest effects are found at moderate intensity and smaller effects are reported at both low and vigorous intensities [12,13,14]. Thirdly, regarding the duration of the physical activity bout, physical activity bouts with durations lasting from 16 to 35 min were used in the majority of the existing literature [14]. Although a minimal threshold of a duration of at least 11 min was reported previously [11], it is important to consider individual preferences in both duration and intensity and that few studies were conducted with ten-minute bouts or less [14]. From a practical point of view, breaks with ten minutes or less might be easier to integrate to the student’s/worker’s day on a regular basis. Fourthly, it is assumed that effects of a physical activity bout on cognition might differ, especially when participants are used to physical activity and show a higher aerobic fitness. There are two diverging proposed directions: Either, individuals with high aerobic fitness might be able to cope with the physiological demands of the physical activity bout more effectively and get larger cognitive benefit from a physical activity bout [14]. Alternatively, ceiling effects in the cognitive assessment might be responsible for a smaller benefit compared with individuals with low aerobic fitness [14]. 

From a behavioral perspective, interventions to influence cognitive function should ideally be conducted not only once or sparsely, but on a regular basis. In this context, both perceived changes of attention [15] and affective states [16] are important to consider. According to the Reasoned Action Approach [15], human behavior (e.g., being physically active to restore cognitive functions) is determined by the intention for a specific behavior (e.g., “It is likely that I become physically active to restore cognitive functions”). One of the key influencing factors on the intention is the attitude towards behavior, which in turn is determined by the beliefs (e.g., “I believe being physically active will restore my cognitive functions”). Consequently, the perception of the individual on the efficacy of an intervention is important to consider. However, cognitive processes are not the only factors influencing human behavior. Opposed to cognitive processes, non-cognitive, affective processes may also play an important role for behavior. The framework of the role of affective states in the maintenance of physical activity behavior lies in hedonic theories [17]. Accordingly, positively influenced affective states (i.e., increased pleasure) during and after a specific behavior (e.g., physical activity) are accompanied with an increased chance to maintain this behavior in the future [17,18].

Following these considerations, the primary goal of the present study was to investigate the effect of a ten-minute physical activity intervention (running) on the cognitive domain of visual attention in the population of physically active sport students. In the context of the literature, we hypothesized that the physical activity bout might show more favorable effects on visual attention compared with a sedentary control condition. As a secondary goal, we were interested in potential accompanying effects of the intervention on perceived attention and affective states, which were previously connected to future behavior. 

## 2. Materials and Methods 

### 2.1. Design and Participants

The present parallel-group randomized controlled trial comprised 51 participants [17 females and 34 males, mean age: 22.3 years (standard deviation: 2.0)] and consisted of an initial screening phase and an experimental phase one week after (Figure 1). The participants were randomly assigned to a physical activity group (*n* = 26) or a sedentary control group (*n* = 25) using block randomization (block size: two participants, allocation ratio: 1:1) and the random number function in Excel (Microsoft, Washington, DC, USA). All data were collected at the identical room at the University of Innsbruck, Department of Sport Science. Detailed information about the procedure was given to the participants prior to the study start, and participation was on a voluntary basis and without consequences in case of non-compliance. All measurements and interventions were conducted according to the Declaration of Helsinki. The project was approved by the Board for Ethical Questions in Science of the University of Innsbruck (50/2019).

All participants were sport science students recruited on a voluntary basis from different lectures in sport science (summer 2017, autumn 2017, and summer 2018). Sport students can be considered as a population with regular physical activity and higher-than-average aerobic fitness. At the University of Innsbruck, all students must pass a comprehensive test for physical fitness (e.g., endurance test: 2900 m in a maximum of twelve minutes for male students) [19]. The exclusion criteria assessed by self-report were acute (in particular cardio-respiratory) sickness and injuries. Allocation sequence, enrolling of participants, and assignment to interventions were done by the first author. No a priori power analysis was performed; however, following the considerations stated recently [14], a sensitivity analysis for the present sample was conducted for the main outcome using G*Power 3.1 (University of Düsseldorf, Düsseldorf, Germany) [20]. Based on the assumptions of alpha = 0.05, power = 0.80, and using a Students *t*-test on independent means as the statistical analysis, an effect size of *d* > 0.8 is revealed as significant with the present sample size of 51 participants. The sample size was comparable to previously conducted studies on acute bouts of physical activity on cognition where a median sample size of around 20 participants within each group is reported [14].

### 2.2. Procedure

The experimental procedure was identical for both groups with the exception of the intervention (Figure 2). Data were collected on three different time points: immediately prior to intervention (pre), immediately after intervention (post), and approximately 30 min after intervention (follow-up). Affective states and perceived attention were measured three times. Visual attention and perceived exertion were assessed once. The time of day at the beginning of the study course was 3:30 p.m. 

### 2.3. Interventions

Participants in the physical activity group were running outdoors together with the test leader and were instructed to choose an intensity between “somewhat hard” and “hard”, which resulted in a distance between 1100 and 1500 m and tapping the lower level of vigorous intensity [21]. Running intensity was assessed using the Ratings of Perceived Exertion post-intervention with respect to the intervention time [22]. The rating scale ranges between 6 (“no exertion”) and 20 (“maximum exertion”). Information about the psychometric properties can be found at Borg [22], and Noble and Robertson [23]. Total time for intervention duration was 10 min; however, leaving and entering the university building was included in this time. For the identical time frame, participants in the control group were instructed to remain in a sedentary position. Reading, talking, and usage of the bathroom was allowed.

### 2.4. Measurements

#### 2.4.1. Initial Screening 

Information on self-reported socio-demographic data, physical activity level, and self-efficacy was collected in a web-based questionnaire with 59 questions (total time to complete: approximately 10 min). Physical activity level was assessed using the short form of the International Physical Activity Questionnaire (IPAQ, http://www.ipaq.ki.se/). The IPAQ asks for the frequency and duration of vigorous, moderate, and walking activity of the last seven days and the sitting time per day using 11 items. Energy expenditure in metabolic equivalent minutes (MET min) is calculated as a total score of physical activity. The IPAQ shows good test-retest reliability (ρ = 0.8) and criterion validity against accelerometer measurements comparable to other self-report measures of physical activity (ρ = 0.3) [24]. Self-efficacy was assessed using the uni-dimensional German Skala zur Allgemeinen Selbstwirksamkeitserwartung [25] in which 10 sentences related to general self-efficacy are rated on a four-point Likert scale ranging from “not true at all” (1) to “very true” (4). All responses are summed up to gain a total self-efficacy score ranging from 10 (low self-efficacy) to 40 (high self-efficacy). The scale shows good internal consistency (Cronbachs alpha > 0.8) as reported in Schwarzer and Jerusalem [25], and an acceptable internal consistency (Cronbachs alpha = 0.74) in the present sample. Moderate correlations to test anxiety were reported (*r* = −0.4) [26]. Self-efficacy was included because of the potential relation to cognition reported in previous literature [27].

#### 2.4.2. Visual Attention

Visual attention was considered the primary outcome of the study and was assessed post-intervention using the group version of a modified trail making test (Zahlen-Verbindungs-Test) [28]. The test is similar to the Trail Making Test (part A only) [29] and provides information about the visual search, scanning, and speed of processing [29]. The instructions of the manual were carefully followed. The participants’ task is to connect circled numbers in a numerical sequence (i.e., 1, 2, 3, etc.) as rapidly as possible in a paper-pencil mode. Four differently numbered sheets of paper are used consecutively with a time frame of 30 s each without breaks in between. The mean value of the four highest numbers reached is calculated and considered as the outcome for visual attention. Higher values indicate higher visual attention. Prior to the test, a short example was given to ensure that the task was fully understood by the participants. The decision to assess visual attention post-intervention only was based on expected effects due to familiarization/practice if assessed pre- and post-intervention. The Zahlen-Verbindungs-Test was selected since visual attention is believed to be an important ability in the university context, since the duration of the test is relatively short and since it provides the possibility to be conducted in a group setting. Good test-retest reliability (*r* = 0.8) and moderate correlation coefficients to other attention/processing speed tests (*r* = 0.6) are reported for students [28]. 

#### 2.4.3. Perceived Attention and Affective States

In addition to the visual attention test, information about perceived attention, affective states, and running intensity were collected during the experimental part. As the secondary outcomes of self-rated perceived attention and affective states were assessed three times using a web-based questionnaire (pre, post, follow-up, Figure 2). Participants rated their perceived attention on a single item question (“Please rate your present attention”) with a virtual slider on a visual analog scale ranging from 0 (“low perceived attention”) to 100 (“high perceived attention”). Two-dimensional single-item scales based on the Circumplex model [30] were used to assess affective states of valence (pleasure) and arousal: The Feeling Scale assesses affective valence using eleven answer possibilities ranging from “very good” (+5) to “very bad” (−5) [31,32]. Arousal was assessed using the Felt Arousal Scale [33], which provides six response options ranging from “low arousal” (1) to “high arousal” (6). Convergent validity values were reported for an affective valence between *r* = 0.4 to 0.9 and for arousal ranging from *r* = 0.5 to 0.7 [34]. 

### 2.5. Statistical Analyses

All statistical analyses were performed using SPSS version 24 (IBM, New York, NY, USA). Preliminary analyses included tests on normal distribution separately for each subgroup using Shapiro-Wilk tests. The main analysis was to assess possible differences in visual attention, perceived attention, affective valence, and arousal between the physical activity group and the control group. The primary outcome of visual attention was assessed once (post-intervention) in both groups. Therefore, a Students *t*-test for independent samples with the between-subject factor group (physical activity group, control group) was used on the dependent variable average reached number of the Zahlen-Verbindungs-Test. The secondary outcomes of perceived attention, affective valence, and arousal, were assessed three times. Therefore, a 2 × 3 mixed analysis of variance (ANOVA) was conducted on each outcome to analyze the effect of the between-subject factor group (physical activity, control), within-subject factor time (pre, post, follow-up), and group by time interaction. Significant interactions between group and time were considered as different changes in the parameters. Pre-planned contrasts using the time point “pre” as the reference category were conducted when a significant interaction was found. The parametric analyses were verified using non-parametric analyses, i.e., Mann-Whitney *U*-Test on the dependent variable’s average reached number of the Zahlen-Verbindungs-Test, and change in secondary outcomes.

Additional analyses that contained two one-factorial analysis of covariance (ANCOVA) were conducted on the primary outcome to assess whether significant differences persisted after controlling for the covariates. The selection of the covariates was based on previous research [27,35]. The ANCOVAs consisted of the between-subject factor group (physical activity group, control group), self-efficacy, and physical activity level separately as the covariates, and of the average reached number of the Zahlen-Verbindungs-Test as the dependent variable. Whenever the assumption of sphericity was not met in the ANOVA or ANCOVA, adjustments after Greenhouse-Geisser were applied. 

Cohens *d* was used as an effect size for the group differences and contrasts with the conventions 0.2 (small), 0.5 (medium), and 0.8 (large) [36]. For all Cohens *d*s, 95% confidence intervals (95% CI) were provided [37]. The significance level was set at α = 0.05 (two-tailed). Unless otherwise stated, data are presented as mean and SD. 

## 3. Results

Demographic data for the control group and the physical activity group are shown in Table 1. Group characteristics were widely similar except for the mean physical activity level, which was higher in the physical activity group. No harmful events and no dropouts were observed over the experimental phase.

### 3.1. Visual Attention

Visual attention (according to the average number in the Zahlen-Verbindungs-Test) was significantly higher in the physical activity group (mean: 47.9, SD: 5.5) compared with the sedentary control group (mean: 43.0, SD: 5.4), *t*(49) = −3.17, *p* = 0.003, *d* = 0.89, 95% CI = [0.31; 1.46]) (Figure 3). 

### 3.2. Perceived Attention and Affective States

Both perceived attention, *F*(2, 90) = 5.22, *p* = 0.007, and arousal, *F*(2, 90) = 22.80, *p* < 0.001, showed a significant group by time interaction, indicating a different change in perceived attention between groups over time (Figure 4). Pre-planned contrasts revealed a significantly larger pre–post increase of perceived attention in the physical activity group compared with the sedentary control group, *F*(1, 45) = 8.45, *p* = 0.006, *d* = 0.87, 95% CI = [0.27; 1.46], while the change in pre-follow-up was not significantly different between groups, *F*(1, 45) = 0.15, *p* = 0.705, *d* = 0.40, 95% CI = [−0.16; 0.97]. Similarly, arousal showed a significantly larger pre–post increase in the physical activity group compared with sedentary control group, *F*(1, 45) = 33.34, *p* < 0.001, *d* = 1.68, 95% CI = [1.02; 2.34], while the change in pre-follow-up was not significantly different between groups, *F*(1, 45) = 0.05, *p* = 0.830, *d* = 0.37, 95% CI = [−0.31; 0.82]. Affective valence did not show a significant group by time interaction, *F*(2, 77.0) = 0.08, *p* = 0.898, *d* = 0.26, 95% CI = [−0.31; 0.82], indicating a similar change in affective valence between groups over time (Figure 4). 

Non-parametric inferential statistics revealed an identical interpretation of the results, i.e., a significant difference between the intervention and control groups in the primary outcome, *z* = −2.72, *p* = 0.007, in the change in perceived attention, *z* = −3.21, *p* = 0.001, and in the change in arousal, *z* = −4.61, *p* < 0.001. The change in affective valence did not significantly differ between the intervention and control groups, *z* = −1.01, *p* = 0.313. 

### 3.3. Covariate-Adjusted Analyses

Adjusting for self-efficacy or the physical activity level did not alter the interpretation of the analysis for visual attention. According to the one-factorial ANCOVA, the difference in visual attention between the physical activity and control groups remained significant after including self-efficacy, *F*(1, 47) = 11.01, *p* = 0.002, or the physical activity level, *F*(1, 46) = 5.20, *p* = 0.027.

## 4. Discussion

The main objective of the present study was to investigate the effects of physical activity on visual attention in comparison with a sedentary control condition during a ten-minute break in the population of sport students. The results suggest that visual attention assessed shortly after the intervention might be positively influenced by physical activity compared with a sedentary control condition. Furthermore, physical activity resulted in a transient enhancement of perceived attention and arousal immediately after the intervention; however, these effects were not evident anymore at 30 min after the intervention. 

### 4.1. Cognition Following Physical Activity

The result of higher visual attention shortly after a physical activity bout compared with a sedentary control condition fits well to previously reported evidence about the short-term effects of a single physical activity bout on cognition [11,12,13]. The positive effects seem to be consistent across various cognitive domains [11,12] as well as the time- (i.e., reaction time or total time for a specific part of the cognitive test) and accuracy-dependent measures of cognition (i.e., number of correct responses or errors) [13]. We used a very short cognitive test, lasting only for two minutes because meta-analytic studies suggest that the greatest enhancement in cognitive functions occurs within a critical time window of 11 to 20 min after the physical activity bout [11,12]. Although it must be considered that no assessment of visual attention prior to the intervention was conducted in the present study, novel findings of the present study are that (a) large-sized positive effects of a physical activity bout on visual attention were evident already after a ten-minute physical activity bout and (b) these positive effects were evident in a population of sport students with above-average aerobic fitness. It was reported previously that short physical activity sessions (i.e., ten minutes and shorter) have a negligible effect on cognition [11]. Our findings suggest that a ten-minute physical activity bout may result in enhanced visual attention compared with a sedentary control condition for the present population of sport students. When interpreting the findings with respect to the duration of the physical activity bout, it is important to consider duration as only one component of the physical activity dose, which must be assessed together with the intensity. An inverted U-shape is suggested for the intensity-cognitive benefits relationship with smaller effects at a (very) light and a vigorous intensity and larger effects at a moderate intensity [14]. The intensity used in the present study must be considered as vigorous physical activity [21]; for this reason, smaller effects had been expected. However, together with a short duration of the physical activity bout, a large-sized effect on cognition was observed. There are at least three possible explanations for this finding: A first explanation is that the difference found is not connected to the intervention, but attributed to baseline differences in the physical activity level. The control group showed a lower physical activity level according to the IPAQ compared with the physical activity group. Therefore, the differences might already have been existent prior to the intervention. Although the physical activity level was added as a covariate in the analysis and still produced significant differences, the group difference must be interpreted with caution. The second explanation is connected to the fact that after the experimental condition, the participants entered the university building, which can be considered as a cool-down phase. Cognitive enhancements following physical activity bouts with vigorous intensity were greater after a cool-down phase of at least one minute [14]. A third explanation is that a short bout of physical activity with a vigorous intensity might result in similar effects as a longer bout of physical activity with a moderate intensity. However, it is important to limit this possible explanation by two aspects: Firstly, the assessment used for cognition in the present study is focusing predominantly on one specific lower-level cognitive domain, namely visual attention [28]. Nevertheless, higher-level processes (e.g., working memory, inhibitory control, cognitive flexibility, planning) rely upon lower-level cognitive, perceptual, and motor processes [38,39]. There is evidence that a higher physical intensity is more beneficial when lower-level cognitive domains are assessed [40]. Secondly, the sample of the present study consisted of sport students, who might be more used to vigorous intensity compared with other populations. 

Our findings further suggest that cognition of individuals with high aerobic fitness may also benefit from a physical activity bout. Thereby, the present results contribute to the knowledge gap on aerobic fitness as a potential moderating variable on the cognitive effects of physical activity bouts [14]. In accordance with the present results, no heterogeneity was reported in the effect of physical activity on cognition between low-fit and high-fit individuals [13]. Similarly, adding the physical activity level (as an indirect indicator of aerobic fitness) as a covariate in the covariance analysis did not change the interpretation of the results, indicating that cognitive benefits might be yielded irrespective of the physical activity level. However, speaking practically, the individual’s aerobic fitness remains important to choose an appropriate dose of physical activity bout [41]. 

Hitherto, potential mechanisms for acute changes in cognitive function after single physical activity bouts are not well understood. Recent evidence from animal and human studies focusing on acute changes suggested several potential mechanisms including increased arousal, increased catecholamines, increased cerebral blood flow, and increased neurotrophic factors (e.g., brain-derived neurotrophic factor (BDNF)) [14,42,43]. Furthermore, the expression of human growth factors, increased blood lactate levels [44], and an activation of the locus coeruleus norepinephrine system might play an important role for an increased attentional state of the brain [14].

### 4.2. Affective States and Perceived Attention

The beneficial effect of a physical activity bout on visual attention compared to the sedentary condition was accompanied by significant short-term changes in the secondary parameters: The participants of the physical activity group reported a larger increase from pre- to post-intervention arousal compared with the sedentary group. Even though the mechanisms behind changes in cognitive function after physical activity bouts remain unclear, increased arousal is regularly mentioned as a possible mechanism behind benefits of physical activity on cognition [12,13,14]. From a physiological perspective, increased arousal resulting in an increased release of catecholamines might trigger the cognitive benefits of physical activity [13]. However, the term arousal is not only referring to physiological aspects, but also to an affective state [30], which was assessed in the present study. The present results of increased (affective) arousal in the physical activity group compared with the control group seem to confirm arousal as a potential mechanism. However, one has to keep in mind that the term arousal is a multifaceted construct and therefore, to gain a more detailed knowledge of the beneficial effects of physical activity on cognition, the different components of arousal such as physiological, cognitive, and affective factors should be considered in future studies [14]. One might assume that arousal might have been different prior to the intervention, potentially biasing the analysis of post-intervention and 30 min after intervention. However, it should be noted that the change in arousal was analyzed on differences between groups, thereby considering the arousal prior to the intervention.

Similar to arousal, perceived attention showed a larger increase from pre- to post-intervention, which disappeared 30 min after intervention, indicating only a short-term effect of physical activity on perceived attention. The results on perceived attention can be interpreted as a confirmation of the result in the visual attention test at a subjective level. The participants did not only show a higher visual attention after the physical activity bout, they also perceived a higher level of attention. Keeping in mind the theory of reasoned action approach [15], this rating of perceived attention is potentially related to the belief in the efficacy of an intervention and believed to positively influence future behavior. In this context, an accompanying increased affective valence due to the intervention would have been desirable [17,18]. However, the change in affective valence was similar across groups, indicating that the physical activity bout did not result in enhanced affective valence post-intervention. Given previous research [45], the lack of positively influenced pleasure was unexpected and might be connected to the higher intensity (between “somewhat hard” and “hard”) of the physical activity bout [46]. Therefore, future studies might consider using a lower intensity of the physical activity bout. Furthermore, the increase in arousal and perceived attention was not evident 30 min after the intervention, suggesting only a short-term effect of the physical activity bout. Since a study course typically lasts longer than 30 min, we conclude that the effects on arousal and perceived attention do not last over a total study course.

### 4.3. Limitations

The following limitations have to be considered when interpreting the findings: Firstly, we did not use physiological outcomes to assess the aerobic fitness level and running intensity due to practical reasons. Assessing maximal aerobic capacity at baseline and heart rate during the interventions would provide more insight in the aerobic fitness level and in the intensity during running and sedentary control conditions. Secondly, we applied a between-subject post-test only comparison assessing visual attention only at a singular time following the experimental condition (physical activity/sedentary control) for each participant. Although the randomization is considered as a process to minimize between-group differences in cognition already prior to intervention, we cannot exclude that participants with higher visual attention were allocated to the physical activity group by chance. Using a crossover study design, where participants engage in both the physical activity and sedentary control condition on separate days would offer the possibility to control for inter-individual variability [14]. However, it should be noted that a crossover design also has some (other) sources of bias (e.g., day-to-day variation in the different conditions, longer observation time including potentially higher dropout rate). In the context of the study design, no visual attention assessment pre-intervention was applied in the present study. Thereby, a decline in visual attention in the control group, in which the participants were allowed to read, talk, and use the bathroom, might similarly explain the results. The behavior in the control intervention is common in breaks between study lessons and might therefore be considered a practically adequate comparator in terms of estimating the effects versus usual behavior. However, potential sources of influence on visual attention in the control intervention (e.g., due to a different amount of social interaction) could not be ruled out in the present study. Adding a visual attention assessment pre-intervention is therefore recommended [14]. Thirdly, the physical activity bout was conducted outdoors. According to the attention restoration theory [47], being outdoors (without being physically active) might already positively influence cognition. The positive effects of the physical activity bout in the present study on both cognition and arousal might have been amplified by the outdoor environment [47,48]. Fourthly, activities of the participants prior to the intervention were standardized 42 min before starting the first assessments. We cannot exclude that longer-term effects (with a duration of more than 42 min) of activities prior to the beginning of the study course might have influenced the results. 

## 5. Conclusions

The present study reports positive effects of vigorous physical activity on visual attention in comparison with a sedentary control condition in the population of sport students. Although the present results have to be interpreted with caution due to baseline differences in the physical activity level, we conclude that a ten-minute running bout conducted in a study course break might be considered a more restorative short-term intervention to enhance visual attention shortly after a break compared with a sedentary control condition. Since visual attention can be considered a basal domain of cognition and is relevant for higher-level cognitive processes, the potential effect on visual attention is of practical importance. Furthermore, the ten-minute running bout resulted in a transient short-term enhancement of perceived attention and arousal. Consequently, the running intervention might help students prior to difficult content to elevate their perceived attention. The running intervention is easy to integrate in breaks between study lessons in sport students and we suggest boosting up research activities for a potential transfer of this approach to office work situations. Future research in this field should consider applying a within-subject design. Furthermore, the inclusion of a group of less physically active students and the comparison of physical activity bouts to other short-term interventions (e.g., power napping, caffeine supplementation) might be taken into account.

## Figures and Tables

**Figure 1 ijerph-17-03678-f001:**
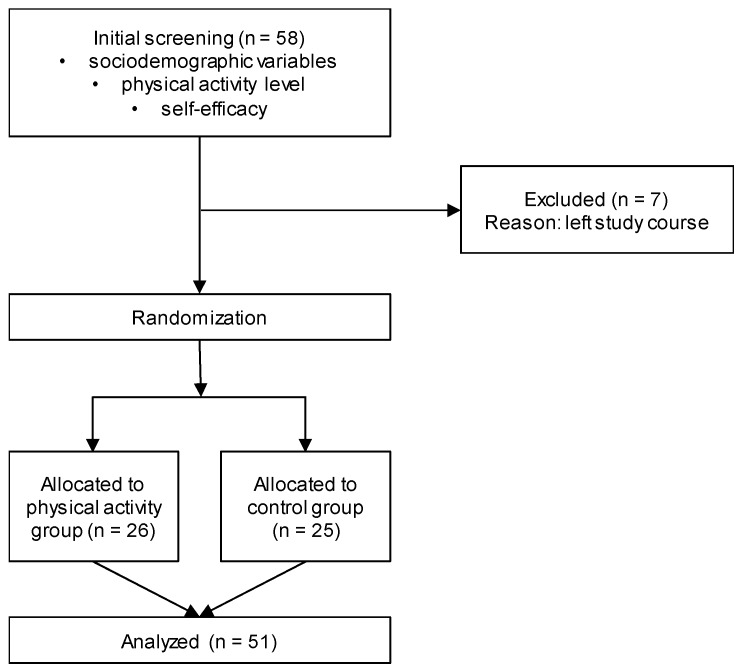
Flow diagram for data collection.

**Figure 2 ijerph-17-03678-f002:**
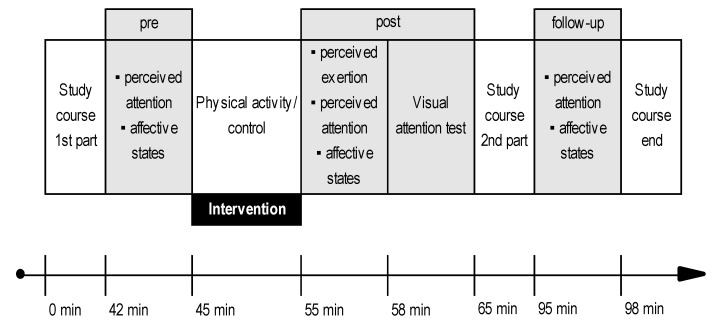
Experimental procedure. Grey fields mark data collection time points.

**Figure 3 ijerph-17-03678-f003:**
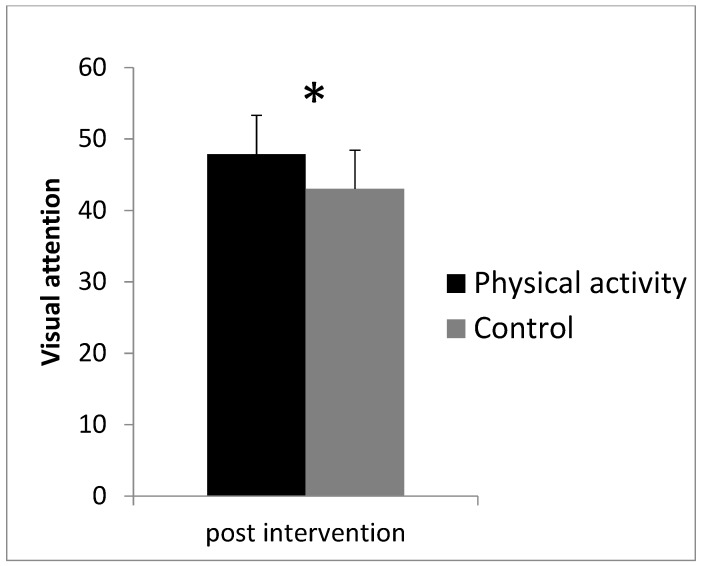
Visual attention (according to the average number in the Zahlen-Verbindungs-Test) separately for each group. Error bars represent standard deviations. * indicates a significant group effect.

**Figure 4 ijerph-17-03678-f004:**
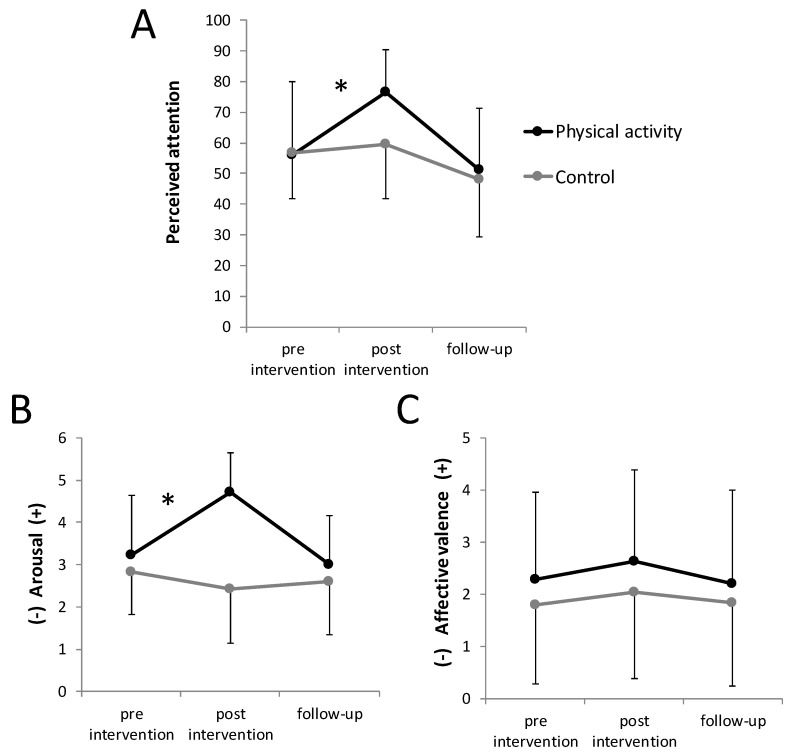
Perceived attention (**A**), arousal (**B**), and affective valence (**C**) over time for each group separately. Error bars represent standard deviations. * indicates a significant group by time interaction.

**Table 1 ijerph-17-03678-t001:** Demographic data of the participants by group (*n* = 51).

Variable	Physical Activity Group (*n* = 26)		Control Group (*n* = 25)	
	Mean	(SD)	Mean	(SD)
Age (years) ^a^	22.8	(2.0)	21.8	(2.0)
Height (cm) ^a^	176.8	(9.5)	174.4	(7.5)
Body Mass (kg) ^a^	71.0	(9.8)	69.0	(10.4)
Body Mass Index (kg/m^2^) ^a^	22.7	(1.9)	22.6	(2.0)
Physical Activity Level (MET min/week) ^b^	7770	(4691)	4461	(2352)
Self-efficacy (10: low, 40: high) ^a^	31	(4)	31	(4)
	**%**	**(*n*)**	**%**	**(*n*)**
Sex, Female ^a^	34.6	(9)	33.3	(8)
Study ^a^				
Sports Science	46.2	(12)	37.5	(9)
Sports Management	38.5	(10)	33.3	(8)
Physical Education	15.4	(4)	29.2	(7)

^a^ missing data *n* = 1 each, ^b^ missing data *n* = 2, MET minutes/week: metabolic equivalent minutes per week.

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
