# Peer review of "Acute Effects of a Short Bout of Physical Activity on Cognitive Function in Sport Students"

_ijerph, 2020, doi:10.3390/ijerph17103678_

Round 1

Reviewer 1 Report

Acute Effects of a Short Bout of Physical Activity on Cognitive Function in Sport Students (ijerph-785446)

The aim of this study was to assess the impact of a single ten-minute physical activity bout on the cognitive domain of visual attention compared to sedentary behavior. This study has interesting practical implications. However, the current manuscript would improve if some issues were addressed.

- In the abstract, it is not clear that the pre-post comparison only was completed for perceived attention and affective states.

- In the introduction, the authors need to explain the relationship that they mentioned between the theory of reasoned action and the measure of perceived attention. They also need to explain why they measured visual attention in this study. How does this measure relate to the type of cognitive activity required of these students in the classroom context?

- Information on internal consistency appears to refer to the original scale. Please, clarify if this information refers to the original scale.

- Did they analyze the responses of men and women?

- The authors need to explain why they did not take pre-post visual attention measures when this factor was considered the primary outcome of the study.

- In the discussion section, it would be interesting to mention whether residual activation or physiological activation following exercise could explain the absence of between-group differences 30 minutes later. To what extent can the benefits of this type of activity last during a class session?

- The main recipients of this type of activity should be students who are not in such good physical shape. Why a group of these characteristics was not included in the study? To what extent are the findings are transferable to other students?

Author Response

Please see the attachment (Reviewer #1).

Reviewer 2 Report

Review report of the original article “Acute Effects of a Short Bout of Physical Activity on Cognitive Functions in Sport Students

The aim of the following study was to investigate how an acute 10 minute bout of aerobic exercise influenced visual attention, perceived attention and affective states compared to a sedentary control. The exercise group showed significantly higher visual attention compared to the control group. Perceived attention and arousal increased significantly from pre to post intervention in the physical activity group but not in the sedentary group.

Overall the study is well written and poses interesting research questions, however I have several major concerns and some minor details to be considered.

Major:

Introduction:

  1. The authors describes well the literature of acute and chronic effects of both sedentary behavior and physical activity. The structure of the introduction could however be improved so that focus will be on the acute effects which this manuscript is about. For example line 44-50 seems to fit to a cross sectional paper not a study investigating the acute effects.

  1. Please describe the literature on the effect of an acute bout of exercise more in depth. Please include timing of the cognitive task as wells as intensity and duration of the exercise bout. Please also justify the choice of the cognitive test used in the present manuscript.

Aim of the research:

  1. The primary outcome of the study was visual attention, however the authors are only measuring that once (after the intervention). This makes it difficult to interpret the differences between the groups as an effect of the intervention. Basically this could just reflect baseline differences between the groups. These differences could potentially be supported by the large difference between groups in physical activity level as seen in table 1 (I know that you adjusted for them in the covariate analysis). In addition, you have no graph on you primary outcome.
  2. Was visual attention the author’s primary outcome when designing the study? If not, then I would suggest to reframe the study so that perceived attention and affective states become the primary outcome and the visual attention can be a seconday.

Material and methods:

  1. Intervention – did the authors obtain any heart rate measurement or likewise? I could be of great advantage for description of the intensity of the exercise bout.
  2. Did the authors consider correcting for multiple testing?
  3.  

Limitations:  

  1. The authors argue that the difference between groups have been minimized due to the randomization, which clearly was not the case when looking at the amount of physical activity between the groups. This again makes it difficult to interpret the primary outcome because this could also just reflect baseline differences. Why I strongly suggest that measures you obtain three times becomes the primary outcomes. This would be easier to justify your design.

Conclusion:

  1. The authors should consider making their conclusion on the data that they have obtained at baseline, after the intervention and at follow-up.

Minor details:

Introduction:

Line 77 – Please elaborate on this statement. It is difficult to interpret

Materials and Methods

Line 97 – One too many parentheses.

Line 111 – The authors are justifying a homogenous sample selection because of the fact that they are sport science students. But clearly this population is not homogenous with respect to the physical activity. Please reconsider this sentence.

Line 190 - Please move the running intensity assessment to the paragraph of the intervention

Line 105 – Please provide a project identification code

Main analyses:

You include both your primary outcome and your secondary outcome in the main analyses paragraph (p. 6, line 229). This is a bit misleading and could preferably be separated.

Author Response

Please see the attachment (Reviewer #2).

Reviewer 3 Report

I would like to thank the authors for this manuscript. Please find my comments and suggestions attached below.

Abstract

It was not clear what stats were used. f it was significant, why only report Cohen's D and not exact p-values?

Introduction

Ln 44 - Incorrect referencing. "The studies summarized in [9]..."

Ln 52-54 - The authors need to distinguish the drop in cognition from a chronic and acute perspective, both of which are likely to have varying mechanisms of action. Long-term reduction in cognition may be related to changes in structure and function associated with the lack of physical activity. However short term changes, may most likely to represent fatigue. Test fatigue, particularly in neuro-cognition research is evident. Similarly paying attention in class is cognitively challenging, which speaks of a different mechanism of action as opposed to long-term sedentary behaviour. 

Ln 61-64 - What is the proposed hypothesis for a small to moderate effect in executive functioning? 

Ln 75-76 - Similar to my previous points, what is the rationale or hypothesis for short bouts of physical exercise to restore cognition? Short bouts do not confer changes to aerobic capacity, therefore there is likely other explanations at play. 

Methods

Ln 97 - change to [standard deviation, SD: 2.0]

Ln 190-191 - Why were no physiological variables collected to provide a subjective measure of intensity, such as heart rate measures? 

Ln 200-201 - Why was an independent t-test done for your primary measure? Please explain?

Ln 210-211 - The description of the statistical analyses for this study is confusing. Perhaps it would be better if the section was revised to state what was done for the primary measure first (i.e. identify and clearly label the between and within-group factors), and then describe what was done with the secondary outcome measures. At the moment, the section is hard to follow. 

Ln 219 - What was the post-hoc test that was used? 

Ln 220 - change to SD instead of standard deviation (SD). 

Results

Ln 235-240 - Your description here may benefit from some clarity. Are you suggesting that there is a significant improvement in perceived attention and arousal from pre-intervention to post intervention OR are you suggesting that the change in the physical activity group is significantly larger than controls?

If it is the former, it means that there are within-group differences but no between-group differences in perceived attention and arousal.

Discussion

Ln 276 - It is unclear why was such a short cognitive test used as a outcome measure. How valid is a 2 minute cognitive task from a practical point of view? Would you still see a difference if the tasks were longer and more complex?

Ln 292-295 - What is/are the proposed mechanism(s) then? Why would allowing a cool down improve cognition? You have provided 2 explanations for your findings, however it does not tell the reader why do you think this is/are the explanations.

Ln 329-336 - Could this increase in arousal be the fact that there is likely to be an increase in heart rate and blood flow? 

Author Response

Please see the attachment (Reviewer #3).

Round 2

Reviewer 2 Report

Thank you for adressing my comments. 

Reviewer 3 Report

The authors have responded to my comments to my satisfaction. No further comments from me.